# Polyelectrolyte Functionalisation of Track Etched Membranes: Towards Charge-Tuneable Adsorber Materials

**DOI:** 10.3390/membranes11070509

**Published:** 2021-07-06

**Authors:** Lisa Wiedenhöft, Mohamed M. A. Elleithy, Mathias Ulbricht, Felix H. Schacher

**Affiliations:** 1Institute of Organic Chemistry and Macromolecular Chemistry (IOMC), Friedrich Schiller University Jena, 07743 Jena, Germany; lisa.volkmann@uni-jena.de; 2Jena Center for Soft Matter (JCSM), Friedrich Schiller University Jena, 07743 Jena, Germany; 3Lehrstuhl für Technische Chemie II, University of Duisburg-Essen, 45141 Essen, Germany; mohamed.elleithy@stud.uni-due.de (M.M.A.E.); mathias.ulbricht@uni-essen.de (M.U.)

**Keywords:** polyampholytes, track-etched membranes, surface modification

## Abstract

Porous adsorber membranes are promising materials for the removal of charged pollutants, such as heavy metal ions or organic dyes as model substances for pharmaceuticals from water. Here, we present the surface grafting of polyethylene terephthalate (PET) track-etched membranes having well defined cylindrical pores of 0.2 or 1 µm diameter with two polyelectrolytes, poly(2-acrylamido glycolic acid) (PAGA) and poly(*N*-acetyl dehydroalanine) (PNADha). The polyelectrolyte functionalised membranes were characterised by changes in wettability and hydraulic permeability in response to the external stimuli pH and the presence of Cu^2+^ ions. The response of the membranes proved to be consistent with functionalisation inside the pores, and the change of grafted polyelectrolyte macro-conformation was due to the reversible protonation or binding of Cu^2+^ ions. Moreover, the adsorption of the model dye methylene blue was studied and quantified. PAGA-grafted membranes showed an adsorption behavior following the Langmuir model for methylene blue.

## 1. Introduction

The pollution of water bodies by heavy metals is an increasing environmental problem driven by industrialization [1]. It directly affects many plants and animals and, by extension, poses a long-term risk for humans due to bioaccumulation. The lack of biological degradability of heavy metal ions leads to their high concentration in creatures higher up on the food chain. This is associated with various health problems, owing to the toxicity and carcinogenetic effects of several heavy metal ions. A second class of substances causing growing concern due to their prevalence in municipal waste waters is pharmaceuticals and personal care products [2]. As with heavy metal ions, these substances can directly interact with the human body, often with undesirable effects. Moreover, conventional water treatment methods are typically not suitable for completely removing these substances [2]. Many pharmaceutically active substances are charged, as are heavy metal ions. Hence, synergistic effects can be exploited for the removal of some of these pollutants [3].

Conventional processes for the removal of heavy metal ions from aqueous solutions rely on precipitation or ion exchange, but are often hampered by the occurrence of contaminated sludges, the lack of selectivity and their inability to be used in a continuous process at high throughput [4]. Low-cost and highly efficient membrane processes could present a viable alternative [5]. An easy to implement method poses the functionalisation of the pore surface of commercially available micro or ultrafiltration membranes with a polymer capable of adsorbing the charged pollutants. In order to prevent leaching of the active polymer, a covalent binding to the membrane via the straightforward approach of in situ polymerisation of the appropriate monomers on the membrane surface seems appealing [6,7].

A promising class of polymers in the context of this application are polyelectrolytes. They are characterised by bearing ionic or ionisable groups in each repetition unit and are thus capable of interacting with charged solutes. Their functional groups dissociate in polar solvents, generating charged polymers and counter ions in solution [8,9,10,11]. Moreover, polyelectrolytes can be distinguished as natural or synthetic, based on their origin. Polyelectrolytes are further classified as polyanions, polycations, or polyzwitterions depending on their charge. A further subdivision is made based on the influence of pH on the charge and charge density. Strong polyelectrolytes are fully ionised over the entire pH range while weak polyelectrolytes reversibly react to changes in pH with varying net charge/degree of ionization [8]. Among weak polyelectrolytes, weak polyzwitterions offer the most versatile platform for tailored responses due to their ability to change between polyanionic, polycationic, and polyzwitterionic states. Yet, this versatility often causes problems concerning solubility and the substances are often only accessible via protected precursors.

One example of such a weak polyzwitterion is polydehydroalanine (PDha), which exhibits an exceptionally high charge density. In recent years, PDha has been used for coatings of magnetic nanoparticles Furthermore, its pH dependant charge inversion was used to adsorb proteins, synthetic polyelectrolytes, and dyes reversibly onto hybrid materials [12,13,14,15,16,17,18]. Typically, PDha is prepared from the precursor poly(*tert*-butoxycarbonylaminomethyl acrylate) (P*t*BAMA) [14,16], but it is also accessible via the free radical polymerisation of *N*-acetyl dehydroalanine (NADha) in aqueous solutions [19]. In addition to being polymerisable in water, PNADha has been shown to be complex and adsorb heavy metal ions even without deprotection [20,21].

Another polyelectrolyte that has been shown to chelate various heavy metal ions depending on the environmental pH is poly(2-acrylamidoglycolic acid) (PAGA). It has been used as a building block for the formation of nanostructured hybrid materials, for water purification, as a cation exchanger, and also for the immobilisation of small molecules [22,23,24,25,26]. In contrast to the polyzwitterionic PDha, it is readily accessible via free radical or RAFT polymerisation of the respective monomer [26]. The structure of the two polyelectrolytes is shown in Scheme 1.

Here we present the surface grafting of PAGA and PNADha onto poly(ethylene terephthalate) (PET) track-etched membranes. Such isoporous membranes had, in earlier studies, been used as versatile model system to evaluate and quantify the effects of pore surface functionalisation on membrane properties including stimuli-responsive permeability or adsorber capacity [27,28]. The polyelectrolyte-functionalised membranes were characterised regarding their surface properties and permeability. Additionally, the membranes were assessed using electron microscopy. Subsequently, the stimuli response and applications for the reversible adsorption of charged species from aqueous solution were investigated. The membrane permeability increases in the presence of acid or Cu^2+^ ions. Moreover, an exposure to Cu^2+^ also alters the surface properties. Using the cationic dye methylene blue as a model substance for small charged organic molecules, the adsorption properties of the membrane were quantified.

## 2. Materials and Methods

### 2.1. Materials

Membranes were purchased from Oxyphen; NADha was purchased from Alfa Aesar; deuterium oxide was purchased from Deutero; and benzophenone, methylene blue, hydrochloric acid, and copper nitrate were obtained from commercial sources. AGA was prepared according to a published protocol [29]. UV-irradiations were carried out using a Hoenle UVACUBE 100 equipped with a 100 W mercury–vapor lamp.

### 2.2. Polymerisation Reactions

In a typical experiment, 0.70 g (5.42 mmol) of NADha and 5 mg (0.03 mmol) of benzophenone are dissolved in 6 mL deionised water via the addition of 3 mL sodium hydroxide solution (5 mol·L^−1^). The pH is then adjusted to the desired value via the dropwise addition of hydrochloric acid (0.5 mol·L^−1^). The solution is deoxygenated by purging with argon for 30 min. Samples of 2 mL volume are polymerised by irradiation in the Hoenle UVACUBE 100 for a specified time (15, 30 or 45 min). The solutions are then neutralised via the addition of concentrated hydrochloric acid. Leftover monomer is precipitated and removed by filtration. The polymer is then isolated via dialysis (3.5 kDa) and freeze-dried.

### 2.3. Membrane Functionalisation

Track-etched membranes were swollen by immersion into 90% (vol.) ethanol in water for a minimum of two hours. Afterwards, they were immersed into an ethanolic benzophenone solution (100 µmol·L^−1^) for an hour in order to absorb the photo initiator. Then, the membranes were allowed to dry via exposure to air for an hour in the dark. Meanwhile, a 10 wt.% solution of the respective monomer in MilliQ water was prepared and degassed by purging with argon for 30 min. The dried membranes were placed in a clear, sealable PE bag and 1000 µL of monomer solution was added. The system was allowed to equilibrate for an hour in the dark without agitation. Afterwards, air and excess monomer solution were removed at room temperature via a rubber roller that simultaneously sealed the bag. The bag was placed between two glass slides and irradiated in the Hoenle UVACUBE 100 for the appropriate time without agitation. The membranes were washed twice with deionised water and stored immersed in deionised water until further use.

The degree of functionalisation was determined gravimetrically based on the mass difference between functionalised and non-functionalised membrane, normalized to the mass of dried pristine membrane, and expressed in wt.% Furthermore, the data were also normalized to the specific surface area of the respective membrane type, obtained by gas adsorption and BET analysis (see below).

### 2.4. Analytical Methods

^1^H NMR spectra were recorded in D_2_O on a Bruker AC 300-MHz spectrometer at 298 K. For calibration, the specific signal of the non-deuterated solvent was used.

Size exclusion chromatography (SEC) measurements in water were performed on a Jasco (Groß-Umstedt, Germany) system equipped with a PU-980 and a RI-2031 Plus refractive index detector 0.08 mol·L^−1^ Na_2_HPO_4_ with 0.05% NaN_3_ (pH 9) was used as a solvent at a flow rate of 1 mL·min^−1^ on a PSS SUPREMA 30 Å at 30 °C. Molecular weight and dispersity were estimated using a calibration with PEO standards.

UV–Vis measurements were performed on an Agilent Cary 60 in a Hellma quartz glass cuvette with a pathlength of 10 mm at room temperature in deionised water or ethanol. The absorbance was measured in a range from 200 nm to 800 nm in 1 nm steps.

Scanning electron microscopy (SEM) was performed on a Gemini 1530 type LEO field emission scanning electron microscope (Carl-Zeiss AG, Oberkochen, Germany). Samples were coated with gold using a Bal-TEC 020 HR Sputtering Coater.

Water permeability measurements were carried out in a stirred ultrafiltration cell (Amicon 8010, Millipore^®^, Merck KGaA, Darmstadt, Germany, effective membrane diameter = 22 mm) connected to a reservoir of deionised water. Transmembrane pressure was controlled in the range between 0.2 and 1 bar using compressed air. Prior to each measurement, the membranes were prepared by flushing the system with the respective feed for five minutes. To investigate the response of the membranes towards pH the reservoir was charged with an aqueous solution of HCl (0.01 mol·L^−1^). The membrane response toward Cu^2+^ ions and regeneration was investigated via the filtration of water after immersing the membrane sample for one hour in either 5 mL aqueous solutions of Cu(NO_3_)_2_ (0.04 mol·L^−1^) or at least 20 mL HCl (0.01 mol·L^−1^), respectively.

In order to evaluate the adsorption of model dye methylene blue, the powdered polymers were dispersed in portions of 50 ± 4 mg in 5 mL of solutions of methylene blue in methanol (0.005 to 0.1 g·L^−1^). The dispersions were left to equilibrate overnight, then the supernatant was passed through a 1 µm glass fibre syringe filter and analysed via UV–Vis spectrometry.

The adsorption of methylene blue onto the functionalised membranes was evaluated by immersing samples of the membranes into 5 mL of aqueous dye solutions of various concentration between 0.015 and 0.098 g·L^−1^ and equilibrated for one hour. Then the membranes were removed from the dye solution, rinsed with deionised water, and placed in 5 mL of hydrochloric acid (0.01 mol·L^−1^) for another hour. All solutions were then analysed using UV–Vis spectrometry. The dye concentrations of all solutions were calculated using the Beer–Lambert law and converted to the absolute mass of dye in the sample volume. In case of the adsorption experiments, the amount adsorbed was calculated as the difference between the initial amount of dye and that remaining after membrane immersion. For the desorption experiments, the amount of dye present in the formerly dye-free sample volume after the immersion of the membrane was taken as the amount desorbed.

Contact angle measurements were performed on a Dataphysics OCA system with deionised water using the sessile-drop method. The drop size was set to 4 µL. Images were recorded immediately after drop collection and analysed using SCAZO software. Before the measurements, the samples were immersed into an aqueous solution of Cu(NO_3_)_2_ (0.04 mol·L^−1^) or deionised water over night and subsequently freeze-dried.

Gas adsorption measurements were performed on a Quantachrome^®^ ASiQwin™ with Krypton at 77 K (Anton Paar Germany GmbH, Ostfildern-Scharnhausen, Germany). Membrane surface areas were calculated for the pristine membranes with 0.2 µm and 1 µm pore sizes using the BET-isotherm.

## 3. Results & Discussion

### 3.1. Evaluation of Photo-Initiated Polymerization of NADha

In a first step, the reactivity of the monomers in a free radical polymerisation was assessed. For AGA, plenty of reports on the free radical polymerisation in aqueous media are available in the literature. For NADha, reports on the polymerisation in aqueous media are scarce and the polymerisation is described as being inhibited [19]. Thus, preliminary experiments concerning the general feasibility and reaction rate of the photo-initiated free radical polymerisation of NADha in aqueous media were performed. The irradiation time was chosen to be rather long in order to mitigate the inhibition of the polymerisation of NADha in aqueous media [19]. The influence of pH and irradiation time was investigated (Appendix A), revealing no significant increase of molecular weight for irradiation times longer than 15 min. In fact, at high pH the molecular weight decreased drastically with the increasing irradiation time, indicating a possible degradation of the polymer. Otherwise, increasing the pH had no significant influence on the outcome of the reaction. A slightly higher molecular weight was observed when methanol was added to the reaction mixture in an attempt to improve the solubility of the photo initiator. Using free radical polymerisation in aqueous media, PNADha was obtained with molecular weights between 23,900 g·mol^−1^ and 47,800 g·mol^−1^ and dispersities between 2.33 and 2.61, as determined using the aqueous SEC measurements (PEO calibration).

### 3.2. Photo-Initiated Graft Copolymerization of Polyelectrolytes to Porous Track-Etched Membranes

Membranes were functionalised with the polyelectrolytes via heterogeneous grafting onto the surface of track-etched membranes. A protocol (cf. Materials and Methods) established in the Ulbricht group and depicted in Scheme 2. was used for the polymerization [6]. The grafting technique exploits differences in hydrophilicity between the aqueous (highly hydrophilic) monomer solution, the moderately hydrophilic membrane material (PET) and the highly hydrophobic PE-bags used as the reaction vessel. In this way, an optimal coverage of the membrane and pore surface by the monomer solution was promoted. The use of the bag further protects the reaction mixture from exposure to atmospheric oxygen during the equilibration and irradiation steps. It is also transparent to UVA irradiation so that the photoinitiator can be activated efficiently.

Initially, the general feasibility of grafting PAGA to the surface of track-etched membranes via UV-initiated surface polymerisation was investigated. A monomer concentration of 10 wt.% and an irradiation time of 10 min were chosen. Using these conditions, membranes with nominal pore sizes of 0.2, 1, and 5 µm could be readily functionalised. The membranes with a 5 µm pore size showed the highest degree of functionalization, with an average increase in weight of 6.7% of total membrane mass, yet the permeability was only reduced by 5% (from 179,000 L·(h·m^2^·bar)^−1^ to 170,000 L·(h·m^2^·bar)^−1^). Both effects can be attributed to the large pore size. While the pores are easily accessible for reagents permitting efficient functionalisation, a thick layer of polyelectrolyte is needed to significantly decrease the pore size and, thus, the membrane permeability. The low effect of grafted polyelectrolyte layers on effective membrane pore sizes for base membranes with a 5 µm pore diameter led to the decision to not perform further evaluations of these membranes.

The results for the membranes with smaller pore sizes were more promising. For the membrane with a pore size of 1 µm, an average functionalisation of 1.2 wt.% PAGA could be achieved, resulting in a 73% decrease in permeability from 14,990 L·(h·m^2^·bar)^−1^ to 4100 L·(h·m^2^·bar)^−1^. In the case of the membrane with a pore size of 0.2 µm, the average PAGA-functionalisation that could be achieved was 4.4 wt.%, even higher than for the membranes with 1 µm pore size. On the other hand, the permeability decreased less strongly, by 47% from 8990 L·(h·m^2^·bar)^−1^ to 4750 L·(h·m^2^·bar)^−1^. The extent of reduction in permeability as well as functionalisation for the different membranes are shown in Figure 1.

After establishing the general feasibility of surface-grafting of PAGA onto PET track-etched membranes, the influence of monomer concentration and irradiation time was studied (Figure 1C). Owing to solubility, the maximum monomer concentration used was 10 wt.% Additionally, the monomer concentration was reduced to 5 wt.% and the irradiation time was varied. Based on the strongly reduced functionalisation and change in permeability for the samples with a monomer concentration of 5 wt.% and an irradiation time of 10 min, the respective experiment with an irradiation time of 5 min was not performed. Increasing the irradiation time to 15 min still could not mitigate the effects of the decreased monomer concentration. Owing to the satisfactory functionalisation and permeability reduction obtained for a monomer concentration of 10 wt.% with an irradiation time of 10 min, the irradiation time was not further increased. A decrease of the reaction time to 5 min caused decreasing functionalisation and permeability reduction, as expected. Based on these results, PAGA-functionalised membranes for the following experiments were prepared using monomer solutions with a concentration of 10 wt.% and an irradiation time of 10 min.

Using the same strategy, the track-etched membranes were also functionalised with PNADha. Here, the irradiation time was increased to 15 min to compensate for the lower reactivity. Independent of the membrane pore size, the sample weight increased, and the hydraulic permeability decreased upon functionalisation with PNADha (Figure 1D). For the degree of functionalisation, the effect was more pronounced for the membrane with a pore size of 0.2 µm. Here, an average increase of 90 µg or 0.8% of total membrane mass could be observed, while for the membrane with a pore size of 1 µm the increase in mass was only 0.03% on average. Concerning the reduction in permeability, the strength of the effects was different. Here, the decrease in permeability was 64% for the 1 µm membrane (from 14,990 L·(h·m^2^·bar)^−1^ to 5370 L·(h·m^2^·bar)^−1^) upon functionalisation with PNADha, while for the membrane with 0.2 µm pore size PNADha functionalisation reduced the permeability by 50% from 8990 L·(h·m^2^·bar)^−1^ to 4460 L·(h·m^2^·bar)^−1^. The fact that PNADha-functionalisation seems less effective but leads to a stronger reduction in permeability compared to the results obtained with PAGA will be discussed below.

In addition to membrane functionalisation with PNADha, deprotection to yield PDha was attempted via acidic hydrolysis [30]. Although the cleavage of the acetyl group could be achieved, the PET membranes were not stable under the reaction conditions. Therefore, no PDha functionalised membranes could be obtained via this pathway and the experiments were discontinued. Instead, the work was focused on PNADha-grafted membranes, as the protected precursor polymer also has the ability to interact with heavy metal ions [20,21].

The influence of functionalisation on the wettability of the membranes was also investigated (Table 1, Appendix A). Upon functionalisation of a membrane with a polyelectrolyte, two distinct effects are expected. On the one hand, the polyelectrolyte should increase the hydrophilicity of the surface (thus decreasing the contact angle). On the other hand, any surface functionalisation itself affects surface composition and structure as well as pore size. This has, in turn, effects on the contact angle. For example, trapped air can cause a high contact angle even with a highly hydrophilic surface [31,32].

For membranes grafted with PNADha, the contact angle increased for both used pore sizes. The increase was stronger for the membrane with the larger pore size of 1 µm. This indicates a small difference in hydrophilicity between PET and PNADha, as well as a larger influence of surface texture. In the case of PAGA-grafting the effect of functionalisation on the contact angle differs with pore size. For the membrane with a pore size of 1 µm, the contact angle increases upon PAGA functionalisation, which we attribute to changes in the surface roughness. For membranes with a pore size of 0.2 µm, the grafting of the polyelectrolyte caused a significant decrease in the contact angle. This is attributed to an increase in surface hydrophilicity as the dominant effect. The reasons for these differences between PAGA and PNADha functionalization will be discussed below.

Moreover, the functionalised membranes were analysed using scanning electron microscopy (SEM), although no significant differences were visible after functionalisation. The micrographs were assessed visually and with the program ImageJ. As shown in Appendix A, all membranes were between 21 and 23 µm in thickness, which was not significantly influenced by grafting a polyelectrolyte onto the membranes. In addition to the membrane thickness, the pore size distributions were analysed. Here, samples with 60–600 pores were evaluated automatically using the particle analyser function of the software ImageJ. No significant difference could be found after polyelectrolyte grafting (Table 1). This lack of difference in pore size and appearance is attributed to the fact that samples are analysed in the dry state and after metal sputtering, so that the thin grafted layers cannot be visualized.

The changes in membrane weight, hydraulic permeability, and contact angle in association with the surface grafting confirmed a successful functionalisation. The extent of functionalisation and the pK_a_ values of the carboxylic groups differ between the two polyelectrolytes. The difference in acidity causes differences in the degree of protonation and, thus, the total charge of the polyelectrolyte chains, which translates into variations in chain extension. Owing to the higher degree of functionalisation achieved with PAGA and the higher acidity of this polyelectrolyte, the decrease in permeability is significantly stronger in the case of membranes with a pore size of 1 µm (Figure 2). The unexpectedly large reduction in permeability found after PNADha-functionalisation suggests additional effects, besides simple chain extension influencing the permeability of polyelectrolyte grafted membranes. Possible explanations are differences in grafting density and chain length or different locations of the grafted polyelectrolytes (i.e., on the top and bottom surfaces vs. inside the pores).

In fact, that PNADha functionalisation is less effective but leads to a stronger reduction in permeability compared to the results obtained with PAGA suggests a different location of the polymer chains. PAGA seems to be located on the top and bottom surfaces as well as inside the pores, while PNADha might be solely located on the inner pore walls. Thus, the effect of a certain degree of functionalisation is more pronounced for PNADha. This consideration also agrees with the changes induced on the contact angle. Functionalisation with PNADha inside the pores leads to a reduced pore diameter and, thus, surface roughness, increasing the contact angle. For functionalisation with PAGA, the strong decrease of the contact angle of the membrane with a pore size of 0.2 µm agrees with the supposed functionalisation on the top and bottom surface, increasing the surface hydrophilicity. In the case of PAGA functionalisation of the membranes with a pore size of 1 µm the higher accessibility of the pores promotes the functionalisation inside the pores over the functionalisation on the top and bottom surfaces, and the effect of the decreased surface roughness is more dominant, thus increasing the contact angle. A possible explanation for the different location of the polyelectrolytes is their different reactivity. While AGA polymerises quickly and readily all over the available membrane surface, the polymerisation of NADha is rather inhibited and might need an acceleration of the reaction rate via the cage effect (i.e., confinement in the pore space).

### 3.3. Stimuli-Responsive Permeability of Polyelectrolyte Grafted Membranes

A different location (i.e., on the outer membrane surface and within the pores, as illustrated in Figure 3 will have additional implications for the response of permeability to stimuli by solution conditions. Even though both arrangements can occur simultaneously, they will be discussed separately to improve clarity. When the polyelectrolyte chains extend into the pores, they decrease the hydraulic permeability of the membrane. When the chains are charged alike, the repulsive forces cause the chains to be extended and block larger parts of the pore volume, thereby decreasing the membrane permeability. When the charges are neutralised, there is no electrostatic repulsion. The polyelectrolyte chains collapse, and a larger proportion of the pore volume becomes accessible allowing an increased permeability. In case the chains are located on the top and bottom surfaces of the membranes, the effects of charge neutralisation are opposite to what is expected for a functionalisation within the pores. When the net charge is not zero, the polyelectrolyte chains are extended due to electrostatic repulsion. However, the chains do not block the interior of the pores. They are only present in dilute polymer state on the PET surface or at the entrance of the pore. When the charges are neutralised, the chains collapse and block access to the pores. Thus, the permeability will decrease upon charge neutralisation. As illustrated by the very high permeability of the membrane in Figure 4A the effect of chain collapse seems to be more pronounced in the case of copper as bivalent cations. In case of the extended chains, there are only few crosslinks possible by the counterions. Yet, previously published results suggest no significant differences between protons and sodium ions [26].

The effect of the stimuli Cu^2+^ ions and pH on the hydraulic permeability of the membranes was assessed. The stimuli induced changes in the membrane permeability, regardless of pore size and type of polyelectrolyte used. The results of the stimuli response tests are summarised in Figure 4.

In case of the membranes with PAGA-functionalisation and a pore size of 0.2 µm (Figure 4A), the permeability was significantly increased in the presence of acid. After the exposure to Cu^2+^, the permeability initially increased strongly, and afterwards quickly decreased to values close to the initial state before the application of any stimulus. This difference can be explained, as in the case of Cu^2+^, wherein the exposure was limited to a single event prior to the measurement. Thus, the cations are partially washed out of the membrane during the measurement, whereas the addition of acid leads to a permanent drop in pH value and, with that, an increased permeability. Yet, the leaking of the Cu^2+^ ions as well as the stability of the permeability after washing the membrane in an acidic solution to regenerate indicated a recyclability of the membranes. The stimuli-responsive behaviour is consistent with a functionalisation within the pores.

For membranes with PAGA functionalisation and a pore size of 1 µm (Figure 4B), a similar but stronger response was observed in the presence of acid. Both PAGA-functionalised membranes showed a comparable initial permeability regardless of the pore size. This indicates the most pronounced chain extension of PAGA, leading to lower permeability. The presence of acid then induced the collapse of the polyelectrolyte chains, allowing a much higher permeability under acidic conditions. Exposure to Cu^2+^ ions seemed to have no effect. Yet, the washing-out of the metal ions, as was observed for the PAGA-functionalised membrane with a 0.2 µm pore size, could be occurring in this case as well. For the membrane with the larger pore size, a higher water permeability is possible with collapsed pores possibly accelerating the leeching of ions. The reverting of the permeability to values observed before the application of any stimuli demonstrates the recyclability of the system. Attempts to regenerate the membrane often resulted in strongly increased values for permeability, possibly indicating some mechanical damage to the membranes caused by the handling.

The effects of exposure to Cu^2+^ and acid on the permeability of PNADha-functionalised membranes with a pore size of 0.2 µm are comparable to those for PAGA-functionalised membranes. Both stimuli led to an increase in membrane permeability, while the membranes can be regenerated to values found before the application of any stimulus (Figure 4C). This behavior is again consistent with a functionalisation inside the pores. The strength of the increase of permeability differs between PAGA and PNADha functionalised membranes. Moreover, the permeability remained increased after exposure to Cu^2+^. These differences could be correlated to differences in pK_a_; Cu^2+^ binding was stronger and therefore leaching was less extensive in case of PNADha.

In response to acid, PNADha-functionalised membranes with a pore size of 1 µm showed an increase in permeability, as did the other functionalised membranes (Figure 4D). Against what was observed for the other membranes, the exposure to Cu^2+^ led to a decrease in permeability. This could possibly be due to the higher Cu^2+^: polymer ratio in this scenario, owing to the lower extent of PNADha functionalisation. The membranes can, however, be regenerated to show a permeability in the range observed before the application of any stimuli.

Generally, the membrane responses are consistent between the polyelectrolytes and the different pore sizes. The main differences were found in the strength of the effects. These differences were attributed to differences in pK_a_ values. The largest exception was found for the response of PNADha functionalised membranes to the presence of Cu^2+^ ions. In this case, the change in permeability was opposite to what was expected and observed for the other membranes. Here, a congestion of the pores is assumed. The responses to the stimuli applied occurred fast, with changes being visible within minutes. In case of the membranes’ reaction towards an acidic environment, the deswelling took place during the change of the filtration feed, even before the first measurement was taken. Apart from these observations, the kinetics of swelling and deswelling were not investigated further. Thus the deswelling of the membrane in response to an exposure to Cu^2+^ ions cannot be accurately timed to values of less than an hour. Additionally, minor differences were observed concerning the regeneration. Except for samples that had been damaged due to handling, the permeability reverted to values in the range observed before stimuli were applied.

### 3.4. Adsorption of Cationic Solutes

In addition to the changes of permeability in response to certain stimuli, the absorption of charged species was investigated. In a first step, the membranes were exposed to Cu^2+^ ions and changes in the contact angle were monitored. In some cases, the contact angle increased after exposure to Cu^2+^. This indicates a decrease in hydrophilicity and/or a decrease in surface roughness upon formation of the polyelectrolyte-Cu^2+^ complexes (Appendix A). In the other cases, a decreasing contact angle was observed, which we tentatively attributed to changes in the surface roughness. The largest difference was found for the membranes with a pore size of 1.0 µm and PNADha functionalisation. Interestingly, small increases could also be observed for the non-functionalised base membrane, which can possibly be assigned to surface hydroxyl groups from the hydrophilization process interacting with the Cu^2+^ ions.

The adsorption of the model dye methylene blue was studied. In a first step, the interaction between the homopolymers and solutions of methylene blue in ethanol was assessed via UV–Vis spectroscopy. For PNADha, adsorption could only be shown for the higher concentrated solution (Appendix A). The results are strongly influenced by scattering which was probably caused by polymer left in dispersion after filtration. In general, both polyelectrolytes exhibited a comparable adsorption of methylene blue from non-aqueous solutions. In aqueous solutions, the adsorption behaviour could additionally be influenced by the degree of protonation of the polyelectrolyte.

Subsequently, the adsorption of methylene blue from aqueous solution to the polyelectrolyte-grafted membranes was investigated. As expected based on the pK_a_ values, the highest adsorption was observed for the PAGA-grafted membranes (pK_a_(PAGA) = 4.32) [26]. The absorption on base membranes and PNADha-functionalised membranes is much lower and little differences are found between these types of membranes (pK_a_(PNADha) = 4.95) [20] (Table 2). This agrees with the lower degree of functionalisation of the PNADha-grafted membranes, and could indicate a low degree of ionisation of the PNADha chains in water at neutral pH. Although this seems to contradict the results for copper adsorption, the adsorption of metal ions is not solely influenced by charge. Complex formation between metal ions and polymers can also be influenced by the nature and geometry of the ligand polymer. For example, a typical binding mode for Cu^2+^ ions is the bridging of two carboxylic acid groups of the same or different macromolecule chains.

In the case of the PAGA-grafted membranes, plotting of the membrane loading against the equilibrium concentration of methylene blue revealed an adsorption behaviour following the Langmuir model (Figure 5B,D). In the case of the other membranes, the extent of adsorption was much lower and did not follow any clear trend (Figure 5A,C). The PNADha-functionalised membrane with a pore size of 0.2 µm showed a slight increase in the amount of methylene blue adsorbed and desorbed with the increasing dye-concentration. In case of the PNADha-functionalised membrane with a pore size 1 µm, the adsorption increased as expected. However, with increasing dye concentration, the desorption decreased. As expected, the changes observed for the base membrane are very small and fluctuate around zero for both adsorption and desorption.

Generally, the influence of the initial dye concentration is pronounced only for the PAGA-grafted membranes. Here, the dye adsorption also takes place to the largest extent. The membranes grafted with PNADha also seem to be effective in adsorbing a low amount of dye. However, the low degree of functionalization, in combination with a low degree of ionization, does not enable the adsorption of larger amounts. Compared to the adsorption to and desorption from the non-functionalised base membranes, the PNADha-functionalised membranes perform within the margin of error (Figure 5A,C).

Fitting the adsorption data based on a Langmuir model revealed comparable parameters for both PAGA-grafted membranes, as expected for surfaces grafted with the same substance. In case of the membrane with a pore size of 0.2 µm a maximum monolayer adsorption of q_mono, max_ = 1.40 ± 0.74 and a Langmuir coefficient of K = 15.95 ± 14.03 mL·mg^−^^1^ were found, while for the membrane with the same functionalisation and a pore size of 1 µm values of q_mono, max_ = 0.86 ± 0.10 and K = 47.20 ± 25.52 mL·mg^−1^ were found. Owing to the large confidence intervals, there is an overlap even though the numbers seem to be very different at first glance.

## 4. Conclusions

Both monomers AGA and NADha could be successfully grafted onto the surface of PET track etched membranes, and the surface functionalisation was verified via changes in membrane weight, permeability, and membrane wettability. In case of PAGA-functionalised membranes, the influence of monomer concentration and irradiation time was investigated and optimised to maximise surface coverage while using the lowest irradiation time and monomer concentration possible.

The functionalised membranes were characterised using SEM, permeability measurements, contact angle measurements, and the adsorption of the model dye methylene blue. In agreement with the reactivity of the respective monomers, the degree of functionalization under similar conditions was higher for the PAGA-grafted membranes. Although this trend is also observed for the changes in permeability, the differences between the differently functionalised membranes were less pronounced. The membranes were further analysed concerning their response towards changes in pH value and the presence of Cu^2+^ ions. Both stimuli caused reversible changes in the membrane permeability consistent with polyelectrolyte functionalisation inside the pores. The presence of Cu^2+^ ions also induced changes in the contact angle of the membranes.

In adsorption experiments using methylene blue, the potential of PAGA-functionalised membranes for removal of the dye from aqueous solution could be demonstrated. The adsorption follows a Langmuir model and the relevant parameters q_mono, max_ and K are comparable for all PAGA-grafted membranes independent of their pore size. In the case of the PNADha-functionalised membranes, no significant differences (if compared to the base membrane) were observed.

Under the conditions used here, PAGA clearly outperformed PNADha. However, PNADha still has the possibility to be deprotected to zwitterionic polydehydroalanine with further application potential (e.g., the absorption of anionic species) [33]. Although our initial experiments concerning deprotection failed due to disintegration of the track-etched PET membranes, under suitable reaction conditions or with base membranes from other polymers this pathway still seems interesting. The targeted porous adsorber membranes could yet be accessible, for example by using alkylazide-functionalized polyethersulfone base membranes in combination with alkyne terminated PtBAMA via the Huisgen addition as a “click” reaction. Both parts of this approach have already been performed successfully, albeit not yet in this specific combination [34,35].

## Data Availability

Not applicable.

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
