# Peer review of "Polyelectrolyte Functionalisation of Track Etched Membranes: Towards Charge-Tuneable Adsorber Materials"

_membranes, 2021, doi:10.3390/membranes11070509_

Round 1

Reviewer 1 Report

The manuscript by Wiedenhöft et al. describes the functionalization of track-etched PET membranes with poly(2-acrylaamido glycolic acid) (PAGA) or poly(N-acetyl dehydroalanine for the swelling and absorption capabilities in the presence of metal ions and upon pH value variation. The membranes were prepared after UV-initiator absorption followed by free radical polymerization of the respective monomers. Reaction parameters were varied for optimizing the amount of polar polymers and influence on water flux properties. The polymers and functionalized membranes have been studied by state-of-the-art polymer analytical tools as well as water flux measurements in the presence of copper ions and varying pH values. In general, the manuscript is straightforward and well-written and from my point of view this work can be suitable for publication in MDPI Membranes principle as is. The authors should address the following minor aspect and suggestions:

  1. The membrane functionalization process, i.e., using a PE bag is somehow special.Could the authors give a clearer description how the membranes were prepared? Is there any convection/stirring? The bags were used in order to prevent sticking to the glass slides upon irradiation? Rubber rolling was applied without temperature? Please specify solution of monomer in water (?) in the following sentence: ‘The dried membranes were placed in a clear, sealable PE bag and 1000 μL of monomer solution were added.‘
  2. Introduction: another efficient way for removing pharmaceuticals in a selective manner could by addressed, as described within the following reviews and references: Funct. Mater. 2016, 26, 3394-3404; Energy Environ. Sci. 2017, 10, 1272-1283; Adv. Funct. Mater. 2021, 31, 2009307.
  3. Page 7: the authors should provide some information within the main text, how molar masses via SEC were determined (solvent, salt additives, standards).
  4. Page 12: while discussing a potential confinement effect on polymerization, the authors should comment on the pore size and the potential hydrodynamic volumes of the respective polymer chains. Is a confinement effect justified in this particular case?
  5. Figure 3: While showing the concept of swelling and influence on polymer conformation, counterions should briefly illustrated or at least commented within the scheme description.
  6. Figure 4: the composed Figure 4 lost resolution. Please change upon revision/acceptance.
  7. Could the authors comment on the speed of responsiveness of the functional membranes (pH? copper addition) upon flux measurements? How long will a deswelling take?

Reviewer 2 Report

Manuscript ID: membranes-1286093; Title: Polyelectrolyte functionalisation of track etched membranes: Towards charge-tuneable adsorber material.

This manuscript entitled " Polyelectrolyte functionalisation of track etched membranes: Towards charge-tuneable adsorber material " by Wiedenhöft et al., is a  research type article, which focuses on the preparation of poly(2-acrylamido glycolic acid) (PAGA) and poly(N-acetyl dehydroalanine) (PNADha) membranes  via surface grafting of polyethylene terephthalate (PET) track-etched.  The mechanism and chemistry of the preparation of the polyelectrolyte functionalized membranes were provided and their application for  the adsorption of the model dye methylene blue was studied and quantified. Many parts of the paper are well written and provided scientifically sound methods. After a robust evaluation of this manuscript, I think this manuscript can be published in Membranes journal.

Author Response

We thank the reviewer for the positive assessment of our manuscript.